# Heavy-light $N + 1$ clusters of two-dimensional fermions

Jules Givois[1]⋆, Andrea Tononi[1,2] and Dmitry S. Petrov[1]

**1** Université Paris-Saclay, CNRS, LPTMS, 91405 Orsay, France
**2** ICFO-Institut de Ciencies Fotoniques, The Barcelona Institute of Science and Technology, Av. Carl Friedrich Gauss 3, 08860 Castelldefels (Barcelona), Spain

⋆ jules.givois@universite-paris-saclay.fr

## Abstract

We study binding of $N$ identical heavy fermions by a light atom in two dimensions assuming zero-range attractive heavy-light interactions. By using the mean-field theory valid for large $N$ we show that the $N + 1$ cluster is bound when the mass ratio exceeds $1.074N^2$. The mean-field theory, being scale invariant in two dimensions, predicts only the shapes of the clusters leaving their sizes and energies undefined. By taking into account beyond-mean-field effects we find closed-form expressions for these quantities. We also discuss differences between the Thomas-Fermi and Hartree-Fock approaches for treating the heavy fermions.

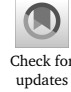

## 1 Introduction

Binding in the fermionic $N + 1$-body model with zero-range interactions is a fundamental problem, necessary for general understanding of fermionic mixtures with mass and population

imbalance. Apart from the interspecies scattering length $a > 0$, which determines the size of the 1+1 cluster, the model is parametrized by the number of heavy fermions $N$, the mass ratio $M/m$, and the space dimension $D$. This parameter space has unexplored spots in spite of the constant interest to the problem from the nuclear-physics side and, more recently, from the ultra-cold-gas community.

In contrast to attractive bosons, which typically always bind, fermionic $N + 1$-clusters bind only above a critical mass ratio, such that the interspecies attraction overcomes the Fermi pressure. Previous studies have shown that binding of larger clusters requires higher mass ratios or lower dimension, which can be explained by the dependence of the Fermi pressure (kinetic energy of noninteracting fermions) on $M$ and $D$. The current status of the fermionic $N + 1$ problem is as follows.

In three dimensions, the $2 + 1$ trimer binds at $M/m = 8.2$ [1], the $3 + 1$ tetramer at $M/m = 8.9$ [2, 3], and the $4 + 1$ pentamer at $M/m = 9.7$ [3]. A qualitative picture which explains the relatively small spread in the critical mass ratios for these bound states, as well as their angular momenta and parities, is that the dimer acts as a $p$-wave-attractive scattering center for heavy atoms and thus accomodates three orbitals with different projections of the angular momentum [3]. These $N + 1$-body systems become Efimovian above certain mass-ratio thresholds in the vicinity of $M/m \approx 13$ and require a three-body, four-body, and five-body parameter, respectively [3–5].

In one dimension there is no Efimov effect, no fermionic sign problem, and no shell effects. The $2 + 1$ trimer exists for any $M/m > 1$ [6] and the mass-ratio thresholds of $N + 1$ clusters steadily grow with $N$ [7]. In the limit of large $N$ their shapes and energies are well described by the mean-field theory; in this limit the critical mass ratios scale as $M/m = \pi^2 N^3/36$ [7].

In two dimensions the atom-dimer attraction in the $p$-wave channel leads to a formation of the $2 + 1$ trimer with unit angular momentum for $M/m > 3.33$ [8]. The $3 + 1$ tetramer emerges almost immediately, for $M/m > 3.38$ [9], pointing to even stronger shell effects than in three dimensions. Exact calculations demonstrate the presence of an excited tetramer for $M/m > 5$ [10] and a ground-state pentamer for $M/m > 5.14$ [9]. Although there is no Efimov effect, the fermionic statistics and rapid growth of the configurational space with $N$ makes the analysis of larger clusters technically difficult.

In this paper we address the large-$N$ limit of the two-dimensional $N + 1$-body problem by applying the mean-field (MF) approximation together with the local-density Thomas-Fermi (TF) assumption for the energy of an ideal Fermi gas. We show that the $N + 1$ cluster emerges for $M/m = 2N^2/C$, where $C = 1.862$. At this critical point the cluster shape is controlled by the nonlinear Schrödinger equation with attractive cubic nonlinearity like in the case of attractive two-dimensional bosons. We thus notice similarities of the critical $N + 1$ cluster with the Townes soliton [11], recently observed in ultracold bosonic atoms [12, 13]. For larger mass ratios our theory gives the shape of the cluster as a function of the parameter $\alpha = 4\pi N^2/(M/m) < 2\pi C$ and the coupling constant for which the solution exists. However, being scale invariant this theory does not predict the size or energy of the cluster. We show that the leading beyond-MF correction for the $N + 1$ cluster with $N \gg 1$ is a local quantity and we use it to calculate the cluster energy and size, which both scale exponentially with $N$. We show that replacing the TF approximation by the full Hartree-Fock treatment is necessary for determining the preexponential factor.

## 2   Mean-field Thomas-Fermi approach for large $N$

We consider a mass-imbalanced fermionic mixture governed by the Hamiltonian

$$\hat{H} = \int \left( -\frac{\hat{\phi}_{\mathbf{r}}^\dagger \nabla_{\mathbf{r}}^2 \hat{\phi}_{\mathbf{r}}}{2m} - \frac{\hat{\Psi}_{\mathbf{r}}^\dagger \nabla_{\mathbf{r}}^2 \hat{\Psi}_{\mathbf{r}}}{2M} + g \hat{\Psi}_{\mathbf{r}}^\dagger \hat{\phi}_{\mathbf{r}}^\dagger \hat{\Psi}_{\mathbf{r}} \hat{\phi}_{\mathbf{r}} \right) d^2 r \,, \tag{1}$$

where $\hat{\phi}_{\mathbf{r}}^\dagger$ and $\hat{\Psi}_{\mathbf{r}}^\dagger$ are the creation operators of light and heavy fermions, respectively, and we set $\hbar = 1$. The short-range heavy-light interaction is characterized by the coupling constant $g = 2\pi/[m_r \ln(2m_r |E_{1+1}|/\kappa^2)] < 0$, where $m_r = mM/(m+M)$ is the reduced mass, $E_{1+1}$ is the dimer energy, and $\kappa$ is the ultraviolet cut-off momentum assumed to be much larger than any other momentum scale in the problem. The dimer energy is related to the heavy-light scattering length by $E_{1+1} = -2e^{-2\gamma_E}/(m_r a^2)$, where $\gamma_E$ is the Euler constant.

We write the MF energy functional for the $N + 1$ system as

$$E = \frac{1}{2m} \int \left[ |\nabla \phi(\mathbf{r})|^2 + \frac{\alpha}{2} n^2(\mathbf{r}) + \gamma n(\mathbf{r}) |\phi(\mathbf{r})|^2 \right] d^2 r \,, \tag{2}$$

where $\phi(\mathbf{r})$ is the wave function of the light atom, the product $N n(\mathbf{r})$ is the density profile of the heavy atoms, and we introduce two dimensionless parameters: $\alpha = 4\pi m N^2/M$ and $\gamma = 2mgN < 0$. Equation (2) is valid for weak interactions, i.e., $m_r |g| \ll 1$. The term $\propto n^2(\mathbf{r})$ is the kinetic energy density of an ideal Fermi gas taken in the TF local-density approximation valid for $N \gg 1$, when $n$ changes slowly on the mean interparticle distance.

To minimize Eq. (2) with the normalization constraints $\int |\phi(\mathbf{r})|^2 d^2 r = 1$ and $\int n(\mathbf{r}) d^2 r = 1$ we introduce the Lagrange multipliers $\epsilon$ and $\mu$ and minimize the grand potential $\Omega = E - \int [\mu N n(\mathbf{r}) + \epsilon |\phi(\mathbf{r})|^2] d^2 r$. The conditions $\delta\Omega/\delta\phi = 0$ and $\delta\Omega/\delta n = 0$ lead to the coupled equations

$$-\nabla^2 \phi(\mathbf{r}) + \gamma n(\mathbf{r}) \phi(\mathbf{r}) = 2m\epsilon \phi(\mathbf{r}) \,, \tag{3}$$

$$n(\mathbf{r}) = -\frac{\gamma}{\alpha} \theta[|\phi(\mathbf{r})|^2 + 2mN\mu/\gamma] \,, \tag{4}$$

where $\theta(x) = (x + |x|)/2$. Equation (3) describes a light atom in an effective well formed by the MF attraction of the heavy fermions. Similarly, Eq. (4) is the TF density profile of the heavy fermions in the MF trap created by the light atom.

The Lagrange multipliers $\epsilon$ and $\mu$ have physical meanings of the energy of the light atom and the chemical potential of the heavy atoms, respectively. For self-bound solutions of Eqs. (3) and (4) these quantities should both be negative since $\phi$ and $n$ are not allowed to spread over the whole space. The binding threshold for the $N + 1$ cluster corresponds to $\mu = 0$, when the heavy atom at the Fermi surface is nearly unbound. In this case Eq. (4) reduces to $n(\mathbf{r}) = -\gamma |\phi(\mathbf{r})|^2/\alpha$. The normalization constraints then imply $-\gamma = \alpha$ and Eq. (3) becomes the nonlinear Schrödinger equation with negative cubic nonlinearity like in the case of attractive two-dimensional bosons. The solution of this problem is the Townes soliton which exists only for a specific value of the coupling constant [11]. In our case, this compatibility condition reads $-\gamma = \alpha = 2\pi C = 11.7$ and the critical wave function is given by $\phi(r) = f(r/R)/(R\sqrt{2\pi C})$, where $f(\rho)$ is the unique nodeless solution of $-f'' - f'/\rho - f^3 = -f$ and $C = \int f^2(\rho)\rho d\rho = 1.862$ [11, 14]. The scale $R$ is arbitrary and cannot be determined from the MF set of Eqs. (2)-(4). Interestingly, the MF energy (2) vanishes for this solution independent of $R$ [15–18]. For bosons this problem is solved by the fact that the renormalized coupling constant $g_r$ depends logarithmically on $R$ [14] leading to a shallow (beyond-MF) minimum in $E(R)$ at a certain $R$. Here, for fermions, the stabilization mechanism is similar, but as we will show below, the beyond-MF contribution has a slightly different form and can be calculated in the local-density approximation.

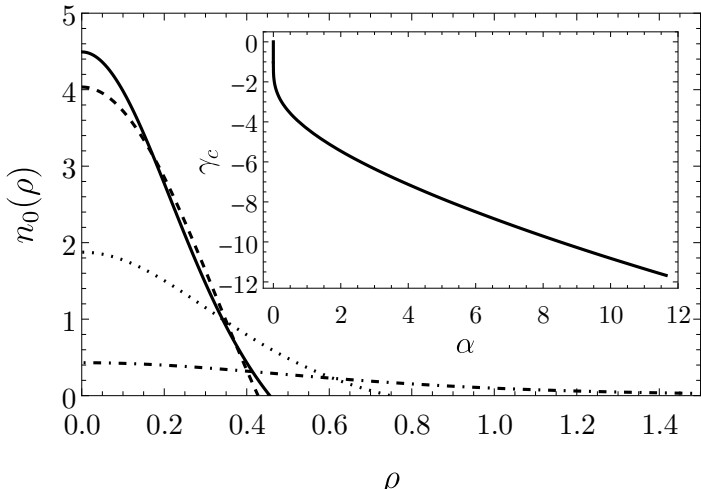

Figure 1: The heavy-atom density $n_0$ versus radius $\rho$ obtained by solving adimensional Eqs. (3) and (4) (with $2m\epsilon = -1$) for $\alpha = 0.8$ (solid), $\alpha = 2$ (dotted), and $\alpha = 11$ (dot-dashed). The dashed curve is the large mass-ratio (or small $\alpha$) limit Eq. (5). The inset shows $\gamma_c$ versus $\alpha$. The small-$\alpha$ asymptote of this curve is $\gamma_c \approx 4\pi/\ln\alpha$ and the end point corresponds to $\gamma_c = -\alpha = -11.7$ and derivative $\gamma'_c = -1/2$.

By numerically solving Eqs. (3) and (4) above the critical mass ratio, i.e., for $\alpha < 2\pi C$, we always find a radially symmetric real nodeless self-bound solution. More precisely, each $\alpha$ gives rise to a family of self-similar solutions which exist only for a certain $\gamma = \gamma_c(\alpha)$. We parametrize this family by the length scale $R = 1/\sqrt{-2m\epsilon}$. Formally setting $\epsilon = \epsilon_0 = -1/(2m)$, Eqs. (3) and (4) become adimensional and we denote their solution by $\phi_0(\rho)$, $n_0(\rho)$, and $\mu_0$. Then, for any $R > 0$ the dimensional solution for the same $\alpha$ and $\gamma = \gamma_c(\alpha)$ reads $\phi_0(r/R)/R$, $n_0(r/R)/R^2$, $\epsilon = -1/(2mR^2)$, and $\mu = \mu_0/R^2$.

All these solutions of Eqs. (3) and (4), for any $\alpha$ and any $R$, correspond to vanishing $E$. Physically, it follows from the fact that Eq. (2) scales with $R$ as $E \propto R^{-2}$. Then, if $E \neq 0$, the system would shrink or expand, contradicting the stationarity of the found solution. That the stationarity of a two-dimensional soliton with cubic (scale-invariant) nonlinearity is equivalent to $E = 0$ has been mathematically shown in Ref. [15] (see also Refs. [16–18]). In our case, to make sure that from Eqs. (3) and (4) indeed follows $E = 0$ we derivate Eq. (3) with respect to $R$ using the scaling properties of $\phi(r)$ and $n(r)$ mentioned above. We then eliminate $\epsilon$ from the result by employing the same Eq. (3) again. We then multiply the resulting equation by $\phi$ and integrate it over space obtaining the equality $\int[-2\phi(r)\nabla^2\phi(r) - \gamma\phi^2(r)rn'(r)]d^2r = 0$, which can further be reduced to the form $E = 0$ with the help of Eq. (4).

We emphasize that $\gamma_c$ is not an external parameter but a characteristic of the solution of the MF Eqs. (3) and (4). On the other hand, $\gamma$ (or $g$) is what we are allowed to tune. Although for $\gamma \neq \gamma_c$ the MF solution is not stationary, it may become stationary once we take into account beyond-MF effects. The final result should not depend on the choice of $g$ and $\kappa$ as long as $a$ is fixed (see Sec. 3).

In Fig. 1 we show $n_0(\rho)$ for a few values of $\alpha$. These profiles are characterized by a finite $\mu_0 < 0$ and, therefore, by a singular behavior of $n_0(\rho)$ at the Thomas-Fermi radius implicitly defined by the equation $\phi_0(\rho_{\text{TF}}) = \sqrt{2mN\mu_0/\gamma}$. The inset in Fig. 1 shows the curve $\gamma_c(\alpha)$.

For large mass ratios Eqs. (3) and (4) can be solved perturbatively. In this regime the heavy atoms are much more localized than the light atom, i.e., $\rho_{\text{TF}} \ll 1$, and the light atom is in the halo state with a small probability to be at $\rho < \rho_{\text{TF}}$. Outside of this region the light-atom wave

function is given by the Bessel function $\phi_0(\rho) \approx K_0(\rho)/\sqrt{\pi}$. In the region $\rho < \rho_{\text{TF}}$ we write $\phi_0(\rho) = \phi_0(0) + \delta\phi_0(\rho)$ and linearize Eqs. (3) and (4) assuming small $\delta\phi_0(\rho) \ll \phi_0(0)$. The linearized equations can be solved straightforwardly leading to the compatibility condition $\gamma_c = 4\pi/\ln\alpha + o(1/\ln\alpha)$ and the density profile

$$n_0(\rho) \approx \frac{4}{\alpha J_1(\sigma_1)\sigma_1} J_0\left(\sqrt{\frac{8\pi}{\alpha}}\rho\right), \ \ \rho < \rho_{\text{TF}} = \sqrt{\frac{\alpha}{8\pi}}\sigma_1, \tag{5}$$

where $J_0$ and $J_1$ are the Bessel functions and $\sigma_1$ is the first zero of $J_0$. The dashed curve for $\alpha = 0.8$ in Fig. 1 corresponds to Eq. (5).

## 3 Beyond-mean-field correction

We see that the MF analysis cannot predict the sizes and the binding energies of the clusters, although it does predict their shapes (up to the rescaling) and determines the threshold mass ratio $M/m = 2N^2/C$. Since the beyond-MF correction is not scale invariant, it introduces preferred length and energy scales, which can be understood from the following arguments. In two dimensions the second-order correction to the energy of two atoms interacting via a delta potential is logarithmically diverging at high momenta. Therefore, the beyond-MF correction to Eq. (2) is dominated by the renormalization of the two-body coupling constant, logarithmic in $\kappa$. It is thus convenient to express the beyond-MF-corrected energy by writing Eq. (2) with $g$ replaced by $g_r = g + \delta g$, where

$$\delta g = -\int_{1/\xi}^{\kappa} \frac{g^2}{k^2/(2m_r)} \frac{d^2k}{(2\pi)^2} = -m_r g^2 \frac{\ln(\kappa\xi)}{\pi}. \tag{6}$$

One can check that the renormalized coupling constant $g_r = g + \delta g$ is cut-off independent up to the second-order terms in the small parameter $m_r|g| \ll 1$. This renormalization removes the cut-off dependence from the energy to this order.

The physical (i.e., cut-off independent) part of the beyond-MF contribution is absorbed into the length scale $\xi$, which is a functional of the fields $\phi$ and $n$, in general nonlocal. Qualitatively, $1/\xi$ is the characteristic momentum governing the many-body or few-body problem at hand. For two atoms in a box $\xi$ is proportional to the box size. For a weakly interacting uniform Bose gas $\xi$ is proportional to the healing length and this result can also be applied in the inhomogeneous case, if the density varies slowly on the scale $\xi$ (see, for instance, [19]). On the other hand, for attractive bosons the local-density approximation does not work since $\xi$ is proportional to the soliton size. However, this very fact that $\xi \propto R$ leads to important predictions for the energy and size scalings of bosonic solitons [14].

In our fermionic $N + 1$ case the typical second-order process contributing to the beyond-MF term is a virtual excitation of the light atom creating a particle-hole excitation in the Fermi sea of heavy atoms. The typical momentum transfer is on the order of the Fermi momentum, which means that $\xi$ is comparable to the mean interparticle separation for the heavy atoms, which scales as $R/\sqrt{N}$. Therefore, the beyond-MF correction in our case is local and can be obtained by analyzing the homogeneous problem. We just need to know the second-order ground-state energy shift for a single light atom immersed in a uniform Fermi sea of heavy atoms with Fermi momentum $p_F$. Having found no answer in the literature we briefly outline this calculation.

Normalizing the single-particle states per unit surface we write the second-order energy correction as [20]

$$\Delta\mathcal{E}^{(2)} = \frac{2Mg^2}{(2\pi)^4}\int \frac{d\mathbf{p}d\mathbf{p}_2}{2\mathbf{p}\mathbf{p}_2 - (1 + M/m)p^2}. \tag{7}$$

The integration domain in Eq. (7) is defined by the inequalities $p_2 < p_F$, $p < \kappa$, and $|\mathbf{p}_2 - \mathbf{p}| > p_F$, corresponding to the following virtual process. The unperturbed state is the impurity at rest and a Fermi sea filled up to $p_F$. The virtually excited state is the light atom at momentum $\mathbf{p}$, a heavy hole at momentum $\mathbf{p}_2$, and a heavy atom at momentum $\mathbf{p}_2 - \mathbf{p}$. We find it convenient to expand $\mathbf{p}_2$ into a vector $\mathbf{p}_\parallel$ parallel to $\mathbf{p}$ and a vector $\mathbf{p}_\perp$ perpendicular to $\mathbf{p}$. The integral over the angle of $\mathbf{p}$ gives $2\pi$. We then integrate Eq. (7) over $p_\perp$, then over $p$, and finally over $p_\parallel$. In this manner, neglecting finite-range corrections $p_F^2 o(p_F/\kappa)$, we obtain

$$\Delta \mathcal{E}^{(2)} = -\frac{m_r g^2 p_F^2}{(2\pi)^2} \ln(\xi\kappa), \tag{8}$$

where

$$\xi = \frac{e^{1/2}}{p_F}\left(\frac{M}{m} + 1\right)\left(\frac{M}{m}\right)^{-1/(1-m/M)}. \tag{9}$$

Since we are interested in the regime $M/m \sim N^2 \gg 1$, Eq. (9) further simplifies to $\xi = e^{1/2}/p_F$ which confirms the qualitative guess $\xi \sim 1/p_F$ and shows that the large mass ratio does not dramatically influence this estimate. We can now proceed with the local-density approximation. Substituting $p_F = \sqrt{4\pi N n(\mathbf{r})}$ into Eqs. (8) and (9), keeping only the leading-order terms at large $M/m$, and going back to notations of Eq. (2) we write the beyond-MF correction to the cluster energy in the form

$$E_{\text{BMF}} = -\frac{1}{2m}\frac{\gamma^2}{2\pi N}\int n(\mathbf{r})|\phi(\mathbf{r})|^2 \ln\frac{e^{1/2}\kappa}{\sqrt{4\pi n(\mathbf{r})N}} d^2 r. \tag{10}$$

Note that $E_{\text{BMF}} \propto N^{-1}\ln N$ is smaller than any of the three terms in Eq. (2), which are $\sim 1$ (for a cluster of unit size). Therefore, Eq. (10) cannot strongly influence the shape of the cluster, but it can remove the degeneracy related to arbitrariness of $R$. Substituting $\phi(r) = \phi_0(r/R)/R$ and $n(r) = n_0(r/R)/R^2$ into Eqs. (2) and (10) and assuming $\gamma = \gamma_c + O(1/N)$ we obtain up to the terms of order $1/N$

$$E + E_{\text{BMF}} = \frac{I_1 \gamma_c^2}{8\pi N m R^2}\left(4\pi N \frac{\gamma - \gamma_c}{\gamma_c^2} + \frac{I_2}{I_1} - \ln\frac{e\kappa^2 R^2}{4\pi N}\right), \tag{11}$$

where $I_1 = \int n_0(\rho)\phi_0^2(\rho) d^2\rho$ and $I_2 = \int n_0(\rho)\phi_0^2(\rho)\ln n_0(\rho) d^2\rho$. Note that up to the chosen accuracy $(\gamma - \gamma_c)/\gamma_c^2 \approx 1/\gamma_c - 1/\gamma$ and $4\pi N/\gamma \approx \ln[4e^{-2\gamma_E}/(a\kappa)^2]$. Minimization of Eq. (11) then gives

$$R_{\min}^2 = \pi N a^2 e^{4\pi N/\gamma_c + I_2/I_1 + 2\gamma_E + O(1/N)} \tag{12}$$

and

$$E_{N+1} = -\frac{I_1\gamma_c^2}{8\pi N m R_{\min}^2} = E_{1+1}\frac{I_1\gamma_c^2}{16\pi^2 N^2}e^{-4\pi N/\gamma_c - I_2/I_1 + O(1/N)}. \tag{13}$$

The parameters $g$ and $\kappa$ drop out from Eqs. (12) and (13) consistent with the fact that the length scale of the problem is given only by the scattering length and the energy scale by the energy of the $1+1$ molecule. The preexponential-factor accuracy in Eqs. (12) and (13) follows from the beyond-MF accuracy of Eq. (11) and from the fact that the weak-interaction parameter $m_r|g| \ll 1$ is equivalent to $1/N \ll 1$ in the self-bound regime since $\gamma \sim 1$. We note, however, that the TF approximation for the kinetic energy of the heavy fermions is guaranteed only to the leading order in $1/N$. To estimate the error we pass to the Hartree-Fock (HF) description.

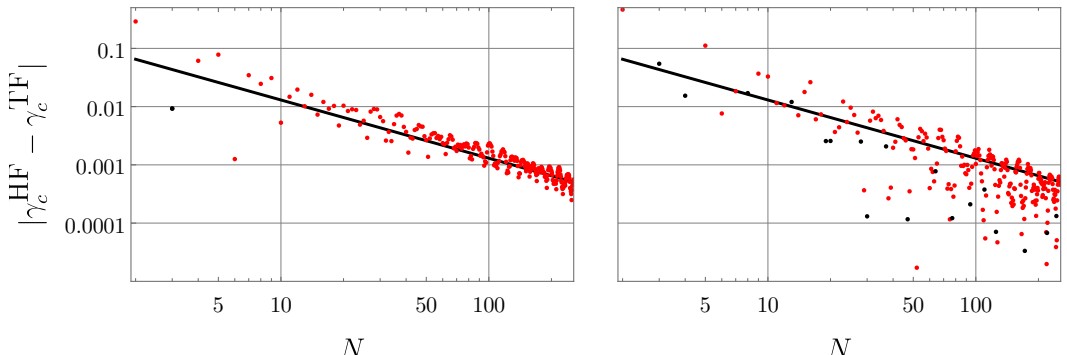

Figure 2: Left panel | $|\gamma_c^{HF} - \gamma_c^{TF}|$ versus $N$ for $\alpha = 2$. Black color corresponds to $\gamma_c^{HF} - \gamma_c^{TF} > 0$, red to $\gamma_c^{HF} - \gamma_c^{TF} < 0$. The solid line marks the slope $\propto N^{-1}$. For reference, for $\alpha = 2$ we have $\gamma_c^{TF} = -5.460$. Right panel | same as the left panel but $\alpha = 8$. Here $\gamma_c^{TF} = -9.714$.

# 4 Hartree-Fock approach

The Hartree-Fock approach for solving the $N + 1$ cluster problem consists in introducing $N$ orthonormal orbitals $\Psi_\nu(\mathbf{r})$ and minimizing Eq. (2), in which the TF approximation $\alpha n^2(\mathbf{r})/(4m)$ is replaced by $\sum_{\nu=1}^N |\nabla \Psi_\nu|^2/(2M)$. With these notations, the minimization of the energy functional with respect to $\phi$ gives Eq. (3) with

$$n(\mathbf{r}) = \sum_{\nu=1}^N |\Psi_\nu|^2/N, \tag{14}$$

and the minimization with respect to the orbitals $\Psi_\nu(\mathbf{r})$ leads to

$$-\nabla^2 \Psi_\nu + \frac{4\pi\gamma N}{\alpha}|\phi|^2 \Psi_\nu = \omega_\nu \Psi_\nu, \tag{15}$$

where $\omega_\nu$ are Lagrange multipliers corresponding to the normalization constraints $\int |\Psi_\nu|^2 d^2r = 1$.

The functions $\phi$ and $n$ determined by the TF and HF approaches are different, but one can easily check that their scaling properties are the same. In the HF method the length scale can thus also get fixed by setting $2m\epsilon_0 = -1$ and denoting the corresponding solutions by $\phi_0$ and $n_0$. In addition, we assume cylindrical symmetry by imposing $\phi_0(\mathbf{r}) = \phi_0(r)$. Equation (15) then splits into one-dimensional Schrödinger equations for functions $\psi_{l,\nu}(r)$, such that $\Psi_\nu(\mathbf{r}) = \psi_{l,\nu}(r)e^{il\varphi}$, with $l$ being the integer angular momentum. States with $l \neq 0$ are doubly degenerate corresponding to $\pm l$. To find the ground state, we diagonalize Eq. (15) and we select the $N$ states with the lowest energy among all possible channels. We then plug these states into Eq. (14), substitute $n_0$ into Eq. (3), and find $\gamma$ for which Eq. (3) has a ground state corresponding to $2m\epsilon_0 = -1$. The function $\phi_0$ is then substituted back into Eq. (15) and the process is repeated until convergence. This procedure results in the critical $\gamma_c^{HF}$ which, in contrast to $\gamma_c^{TF}$ (we use superscripts to specify the method), does not only depend on $\alpha$ but also on $N$.

Figure 2 shows the convergence of the HF value $\gamma_c^{HF}(\alpha, N)$ towards the TF value $\gamma_c^{TF}(\alpha)$ at large $N$ for $\alpha = 2$ and $\alpha = 8$.

We show results up to $N = 256$. The black straight lines indicate the slope $N^{-1}$. We believe that stronger fluctuations for larger $\alpha$ are due to the fact that the uppermost filled heavy-atom

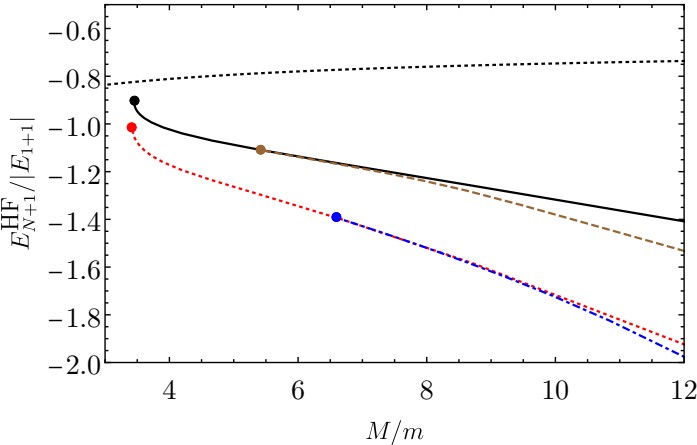

Figure 3: Hartree-Fock energies of the $1+1$ dimer (dotted black), $2+1$ trimer (solid black), $3+1$ tetramer (dotted red), $4+1$ pentamer (dot-dashed blue), and the excited $3+1$ tetramer (dashed brown) in units of the exact dimer energy as a function of the mass ratio. We use the same color code as in Fig. 1 of Ref. [9].

orbitals are closer to the dissociation threshold and the system is thus more sensitive to changes in $N$. Accordingly, we find that the case of larger $\alpha$ and $N$ requires finer and larger spatial grids for accurate calculations.

The fact that $|\gamma_c^{HF} - \gamma_c^{TF}|$ scales as $N^{-1}$ on average shows phenomenologically that to keep up with the claimed accuracy (up to the preexponential factor) for the cluster energy we should use $\gamma_c^{HF}$ instead of $\gamma_c$ in Eqs. (12) and (13). The best TF-based prediction for the cluster energy is therefore $E_{N+1} = -a^{-2}e^{-4\pi N/\gamma_c - 2\ln N + O(N^0)}$. Advantages of the TF approximation is that it has $\alpha$ as the only input parameter and that limiting cases can be worked out analytically. By contrast, the higher accuracy of the HF method comes at the price of doing separate calculations for each $N$ and $M/m$. However, the HF method predicts the structure of the cluster, its angular momentum and parity. It can also naturally handle excited states.

## 5 Hartree-Fock method applied to small clusters

Although the HF method is valid for $N \gg 1$, it is tempting to benchmark its performance for the few lowest-order clusters, for which exact results are known [8–10]. In addition, the obtained solutions can also be used as guiding functions for the fixed-node Monte-Carlo scheme (see, for instance, [21]). We find that the HF method describes these small clusters rather well and we expect the accuracy to further improve with increasing $N$.

As we describe in Sec. 4, iteratively solving Eqs. (3), (14) and (15) we obtain the critical $\gamma_c^{HF}$ and the fields $n_0(r)$ and $\phi_0(r)$. From there we calculate the integrals $I_1 = \int n_0(r)\phi_0^2(r)d^2r$ and $I_2 = \int n_0(r)\phi_0^2(r)\ln n_0(r)d^2r$ and determine the energy from Eqs. (12) and (13). Explicitly,

$$E_{N+1} = -\frac{I_1(\gamma_c^{HF})^2}{8\pi^2 N^2 ma^2}e^{-4\pi N/\gamma_c^{HF} - I_2/I_1 - 2\gamma_E}. \tag{16}$$

We note that although the beyond-MF correction Eq. (10) is obtained within the TF framework, it is sufficiently precise to be used in combination with the Hartree-Fock Eqs. (3), (14), and (15).

In Fig. 3 we plot the energies $E_{N+1}^{\mathrm{HF}}$ in units of the exact dimer energy as a function of the mass ratio. The different curves stand for the $1+1$ cluster (dotted black, in the HF description the heavy atom occupies the lowest $s$-wave orbital), $2+1$ trimer (solid black, occupied are the lowest $s$-wave and one of the two degenerate $l = \pm 1$ orbitals), $3+1$ ground tetramer (dotted red, occupied are the lowest $s$-wave and both lowest $p$-wave orbitals), $4+1$ pentamer (dash-dotted blue, occupied the lowest and the first excited $s$-wave and both $p$-wave orbitals), and the excited $3+1$ tetramer found in Ref. [10] (dashed brown, occupied are the lowest and the first excited $s$-wave and one of the lowest $p$-wave orbitals). One can see that the HF approach reproduces the structure of the levels rather well (cf. [9]), although the artifacts of the approach are also visible. For instance, the trimer and the tetramer emerge immediately with finite binding energies, which is a consequence of the nonlinearity of the equations. The threshold behavior depends on the angular momentum of the orbital or, more precisely, on the convergence properties of the corresponding normalization integral. Since at zero energy the orbitals with angular momentum $l$ behave as $\psi_{l,\nu} \propto r^{-|l|}$, the normalization integral for $s$-wave orbitals diverges. Therefore, a heavy atom in the $s$-wave orbital is in the halo state and does not influence the core. The crossing is therefore smooth (see the crossings of the pentamer and the excited tetramer). By contrast, for $|l| > 1$ the zero-energy orbital function is normalizable meaning that the newly bound heavy atom is inside the core right at the threshold. This creates artifacts due to the nonlinearity. We find that the case $|l| = 1$, in spite of the logarithmic divergence of the normalization integral, is also prone to this nonlinear effect.

The cylindrical-symmetry assumption has to be carefully checked, but it requires a more involved two-dimensional analysis. We leave this task as well as the investigation of higher-order clusters to the future.

# 6 Conclusion

A two-dimensional fermionic $N + 1$ cluster binds for sufficiently large $M/m$. The MF theory valid for large $N$ predicts the threshold value $M/m = 2N^2/C = 1.074 N^2$ and the cluster shape at this point and for larger $M/m$. The beyond-MF analysis based on the local-density approximation gives closed-form expressions for the size and energy of the cluster. The accuracy and practical relevance of the obtained results can be increased by switching to the Hartree-Fock form of the MF density functional. Finally, our findings have implications for ultracold fermionic mixtures. We can think of strongly mass-imbalanced mixtures of $^6$Li with $^{173}$Yb [22, 23] or with other heavy Lanthanides such as Dy or Er.

# Acknowledgments

**Funding information** We acknowledge support from ANR Grant Droplets No. ANR-19-CE30-0003-02 and from EU Quantum Flagship (PASQuanS2.1, 101113690). ICFO group acknowledges support from: Europea Research Council AdG NOQIA; MCIN/ AEI (PGC2018-0910.13039/501100011033, EX2019-000910-S/10.13039/501100011033, Plan National FIDEUA PID2019-106901GB-I00, Plan National STAMEENA PID2022-139099NB, I00, project funded by MCIN/AEI/10.13039/501100011033 and by the "European Union NextGenerationEU/PRTR" (PRTR-C17.I1), FPI); QUANTERA MAQS PCI2019-111828-2); QUANTERA DYNAMITE PCI2022-132919, QuantERA II Programme co-funded by European Union's Horizon 2020 program under Grant Agreement No 101017733); Ministry for Digital Transformation and of Civil Service of the Spanish Government through the QUANTUM ENIA project call -



Quantum Spain project, and by the European Union through the Recovery, Transformation and Resilience Plan - NextGenerationEU within the framework of the Digital Spain 2026 Agenda; Fundació Cellex; Fundació Mir-Puig; Generalitat de Catalunya (European Social Fund FEDER and CERCA program, AGAUR Grant No. 2021 SGR 01452, QuantumCAT U16-011424, co-funded by ERDF Operational Program of Catalonia 2014-2020); Barcelona Supercomputing Center MareNostrum (FI-2023-1-0013); Funded by the European Union. Views and opinions expressed are however those of the author(s) only and do not necessarily reflect those of the European Union, European Commission, European Climate, Infrastructure and Environment Executive Agency (CINEA), or any other granting authority. Neither the European Union nor any granting authority can be held responsible for them (EU Quantum Flagship PASQuanS2.1, 101113690, EU Horizon 2020 FET-OPEN OPTOlogic, Grant No 899794), EU Horizon Europe Program (This project has received funding from the European Union's Horizon Europe research and innovation program under grant agreement No 101080086 NeQSTGrant Agreement 101080086 — NeQST); ICFO Internal "QuantumGaudi" project; European Union's Horizon 2020 program under the Marie Sklodowska-Curie grant agreement No 847648; "La Caixa" Junior Leaders fellowships, La Caixa" Foundation (ID 100010434): CF/BQ/PR23/11980043.

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
