# Peer review of "Heavy-light $N+1$ clusters of two-dimensional fermions"

_SciPost Physics, doi:SciPost Phys. 17, 050 (2024)_

## Round 1 · Referee Report · Anonymous (Referee 1) · 2024-5-12

Report

This paper studies the large-N limit of systems of N heavy identical fermions bound (with zero-range interactions) by a light atom in two dimensions. Such N+1 clusters only bind above a critical mass ratio where the interspecies attraction overcomes the Fermi pressure. By using a mean-field theory, together with a Thomas-Fermi approximation for the kinetic energy of the heavy atoms, the authors determine this critical ratio. Since the mean-field theory is scale invariant, it can additionally predict the shapes of the clusters (up to a rescaling), but not their absolute sizes and energies. In order to obtain the latter, they execute a beyond-mean-field analysis based on the local-density approximation — first by treating the heavy fermions with a Thomas-Fermi approach, and then to improve the accuracy, with a Hartree-Fock approach. They discuss the relative merits of both methods. Last, they apply the (many-body) Hartree-Fock technique to small clusters for which exact solutions are known and find that it performs quite well. The literature is nicely reviewed in the introduction. It is hence evident that this work probes an unexplored region of the parameter space of fermionic (N+1)-body systems with zero-range interactions. The results are also relevant to experiments on ultracold Fermi gases with large spin and mass imbalances (e.g., mixtures of Li-6 and Yb-173). The writing is clear and the methods are detailed. While there is scope for future investigation, this paper tells a complete story. Therefore, I do not believe that anything further needs to be done to this submission, and in my opinion, it is suitable for publication in SciPost Physics.

Recommendation

Publish (easily meets expectations and criteria for this Journal; among top 50%)

---

## Round 1 · Referee Report · Anonymous (Referee 2) · 2024-5-16

Strengths

1-Interesting subject for a broad audience not yet considered in the litterature;
2-Relevant analysis with comparisons between different approaches;
3-Can be used for further studies in this domain

Weaknesses

Even if of good quality, 1- the presentation can be improved 2- the question concerning the numerics can be more detailed

(see report for the details of these two issues)

Report

The physics of one impurity interacting resonantly with N identical fermions is a subject of general interest in the many body problem. With this study in two spatial dimensions, the authors explore a configuration which was not yet solved. After an introduction with an overview of known results, the authors derive the mean field approach (TF) which gives the shape of the density profile but does not fix the spatial scale. This first approach permits one to exhibit the two important dimensionless parameters : the parameter alpha proportional to N^2 over the mass ratio and the second, gamma proportional to the coupling contant g times N where g<0. The system can bind only when alpha is less than a critical value and for a given value of alpha in this interval the mean field equation gives a value of gamma. Next, they use a beyond mean field correction in the local density approximation to determine this unkown scale by showing that in a second order perturbation theory, the coupling constant is a function of the Fermi momentum and of the dimer energy. They thus obtain the cluster energy as a function of the dimer energy. To get more accuracy they use a Hartree-Fock (HF) approach and compare the results obtained numerically with the analytical results of the TF approach. Finally, they compare the HF results with few-body exact results obtained for small N.

The manuscript gives interesting results in the large N limit where exact few-body techniques cannot give any answer. Nevertheless, from my point of view, it can be improved to have more impact.

1) In the introduction, I suggest to explicitly write that the issue studied here concerns the binding of N+1 particles in presence of a shallow dimer (whatever the dimension D) corresponding to the limit of a large and positive scattering length. This is more precise an clear that saying that the interaction is attractive (cf the renormalization or regularization of a delta interaction which means that g delta itself is ill-defined)

2) Part 2: the introduction of the formula of g function of E(1+1) and the UV cut-off kappa is not useful and can bring confusion to the reader: for each alpha there is a gamma and this suggests that kappa is fixed by alpha... However this is not a good reasoning and shows the limitation of the use of a first order perturbation theory.Instead, I suggest to say that writting g delta in the functionnal where g is a given negative constant is a first guess for treating the interaction. Except that, all the analysis is interesting. Perhaps it is valuable to recall that the scale invariance is expected to be broken similarly to what happens in 2D Bosonic systems with a contact force (cf Pitaevskii Rosch collective modes) and it is thus necessary to treat the interaction at the second order of the perturbation theory.

3) Part 3: It is important to emphasize that the shallow dimer energy is the relevant scale: why not express E_{N+1} in terms of E_{1+1} ?

4) Part 4: The dispersion of the points in the right panel of Fig 2 is puzzling:

a) Why not plotting the relative dispersion Delta gamma/gamma (also more relevant) ? the dispersion will be reduced

b) Even with (a) I guess that there will be a larger dispersion for alpha=8 than for alpha=2. Nevertheless, the explanations given are not clear: there are two possibilities as suggested in the text. If the dispersion is due to the vicinity of the threshold, this a very interesting effect which deserves further future studies. If this is a numerical effect, this is less interesting but this can be tested by changing the mesh size and/or the interval of integration. Thus, more infiormations are needed in the text for the numerical analysis (grid used : logarithmic, linear ? , mesh size, interval) and at least vary these parameters would give indications if this is a purely numerical effect.

5) Part 5: In the figure, I suggest to replace E^{exact}{1+1} which appears no where else in the manuscript by E

Except these remarks/suggestions of possible improvments, I think that the paper meets the criteria to be published in Scipost

Requested changes

see report

Recommendation

Ask for minor revision

---

## Round 2 · Referee Report · Anonymous (Referee 1) · 2024-5-30

Report

I am satisfied with the changes made in version two of this submission and thus recommend it for publication.

Recommendation

Publish (easily meets expectations and criteria for this Journal; among top 50%)

---

## Round 2 · Referee Report · Anonymous (Referee 2) · 2024-6-14

Report

The modifications implemented by the authors in the the manuscript make this new version ready for publication in Sci Post . No change is needed.

Recommendation

Publish (surpasses expectations and criteria for this Journal; among top 10%)

---

## Round 2 · Author Response

Dear Editors,

We thank the reviewers for their careful reading of our manuscript, for their positive opinion, and for constructive comments. The first referee has no points to clarify. Below is our response to the second reviewer and the summary of changes. We hope that our manuscript is now suitable for publication.

Sincerely, Jules Givois, Andrea Tononi, and Dmitry Petrov

Referee: 1) In the introduction, I suggest to explicitly write that the issue studied here concerns the binding of N+1 particles in presence of a shallow dimer (whatever the dimension D) corresponding to the limit of a large and positive scattering length. This is more precise an clear that saying that the interaction is attractive (cf the renormalization or regularization of a delta interaction which means that g delta itself is ill-defined)

Response: In the first paragraph of the revised version we now explicitly mention that the scattering length is positive and that it determines the size of the 1+1 cluster.

It is also specified in the abstract and in the first phrase of the introduction that we are dealing with zero-range interactions. Therefore, mentioning that the dimer is shallow may be confusing. The paper is already quite technical and we would not like to touch effective-range effects. When we introduce the cutoff kappa we say that it should be much larger than any other momentum scale in the problem. In particular, it is larger than the inverse size of the dimer.

Referee: 2) Part 2: the introduction of the formula of g function of E(1+1) and the UV cut-off kappa is not useful and can bring confusion to the reader: for each alpha there is a gamma and this suggests that kappa is fixed by alpha... However this is not a good reasoning and shows the limitation of the use of a first order perturbation theory.Instead, I suggest to say that writting g delta in the functionnal where g is a given negative constant is a first guess for treating the interaction. Except that, all the analysis is interesting.

Response: The Referee does not point to any error and the question is semantic. In our formulation g is an auxiliary quantity, which drops out of the final answer. However, it has a well-defined physical meaning. g is the depth of the interaction potential in momentum space and kappa is the corresponding cutoff. We follow the standard procedure of replacing the zero-range boundary condition (governed by the scattering length) by an effective short-range potential (governed by g and kappa). The formula relating these quantities is well known. In contrast to what the Referee says, this formula is useful for us. By developing the perturbation theory for small g up to the second order we obtain results in the g-independent and kappa-independent form, i.e., only in terms of a. This is a very standard way of perturbatively treating two-dimensional systems with weak short-range interactions. The condition of weak interaction allows us to choose a sufficiently large kappa such that one can claim the validity of the result in the zero-range limit.

Thinking of the Referee's comment we guess that the confusion may be due to the difference between gamma_c and gamma which we do not sufficiently emphasize. gamma is an external parameter, whereas gamma_c is not. These two quantities are allowed to deviate from each other. In the new version we add a paragraph in Sec.2 where we explain this. We hope that this solves the issue.

Referee: Perhaps it is valuable to recall that the scale invariance is expected to be broken similarly to what happens in 2D Bosonic systems with a contact force (cf Pitaevskii Rosch collective modes) and it is thus necessary to treat the interaction at the second order of the perturbation theory.

Response: In Sec.3 we mention that the scaling invariance is broken by beyond-MF effects citing the work of Hammer and Son Ref.[14]. As far as we know, Pitaevskii and Rosch in Ref.[17] did not mention this mechanism of symmetry breaking, but we refer to their work in Sec.2 in another context.

Referee: 3) Part 3: It is important to emphasize that the shallow dimer energy is the relevant scale: why not express E_{N+1} in terms of E_{1+1} ?

Response: We added this expression to Eq.(13) and also note that the result is independent of the cutoff, consistent with the fact that the dimer size a is the only length scale in the problem.

Referee: 4) Part 4: The dispersion of the points in the right panel of Fig 2 is puzzling:

a) Why not plotting the relative dispersion Delta gamma/gamma (also more relevant) ? the dispersion will be reduced

Response: gamma is a dimensionless quantity. It is not obvious for us that dividing by gamma brings in more information. In any case, on the log scale this would just shift all the data points in the vertical direction without changing their dispersion. We also note that the values of gamma are given in the caption.

Referee: b) Even with (a) I guess that there will be a larger dispersion for alpha=8 than for alpha=2. Nevertheless, the explanations given are not clear: there are two possibilities as suggested in the text. If the dispersion is due to the vicinity of the threshold, this a very interesting effect which deserves further future studies. If this is a numerical effect, this is less interesting but this can be tested by changing the mesh size and/or the interval of integration. Thus, more infiormations are needed in the text for the numerical analysis (grid used : logarithmic, linear ? , mesh size, interval) and at least vary these parameters would give indications if this is a purely numerical effect.

Response: The numerical procedure is described after Eq.(15). It amounts to solving two coupled one-dimensional (we assume cylindrical symmetry) differential equations, which is a rather easy task by current standards. The grid in the radial direction contains a few thousands of points in the region of interest (inside and around the cluster). We have varied the grid parameters and we are confident that the numerical uncertainty is negligible on the scale of Fig.2. The dispersion is not a numerical artifact. We agree with the Referee that this is an interesting phenomenon for future studies.

Referee: 5) Part 5: In the figure, I suggest to replace E^{exact}{1+1} which appears no where else in the manuscript by E

Response: Done.

---

## Round 2 · List of Changes

1) On page 1 in the Introduction we mention that a is positive and it determines the size of the 1+1 cluster.

2) On page 4 we add a paragraph explaining the difference between gamma_c and gamma. We also mention that the final result of the calculation (after taking into account the beyond-MF effects) should not depend on the choice of the pair g and kappa.

3) In Eq.(13) we added the expression for the energy of the N+1 cluster in terms of the energy of the 1+1 cluster. Right after this equation we point out that g and kappa drop out of the final answer.

4) We removed the superscript "exact" from the vertical label in Fig. 3.

---

## Editorial Decision

published